# Molecular Epidemiological Survey of *Cryptosporidium* in *Ochotona curzoniae* and *Bos grunniens* of Zoige County, Sichuan Province

**DOI:** 10.3390/ani15142140

**Published:** 2025-07-19

**Authors:** Tian-Cai Tang, Ri-Hong Jike, Liang-Quan Zhu, Chao-Xi Chen, Li-Li Hao

**Affiliations:** 1College of Animal Husbandry and Veterinary Medicine, Southwest Minzu University, Chengdu 610041, China; tiancaitang@126.com (T.-C.T.); chaoxi8832@163.com (C.-X.C.); 2Faculty of Agriculture, Forestry and Food Engineering, Yibin University, Yibin 644000, China; 3Agricultural and Rural Bureau of Liangshan Yi Autonomous Prefecture of Sichuan Province, Liangshan 615000, China; 18788911610@163.com; 4China Institute of Veterinary Drug Control (IVDC), Beijing 100081, China; zhuliangquan_ivdc@126.com

**Keywords:** *Ochotona curzoniae*, *Bos grunniens*, *Cryptosporidium*, *SSU rRNA*, epidemiological survey

## Abstract

*Cryptosporidium* is an important zoonotic parasite that infects a wide range of animals, causing diarrhea, growth retardation, and economic losses in livestock. Although *Cryptosporidium* has been reported in some regions of Aba Prefecture, there is limited research on the prevalence and distribution of *Cryptosporidium* spp. in *Ochotona curzoniae* (plateau pika) and *Bos grunniens* (yak) in Zoige County. Therefore, the current study aimed to investigate the species and genotypes of *Cryptosporidium* in *O. curzoniae* and *B. grunniens*, and to reveal their potential zoonotic risk in Zoige County of Sichuan Province, China. The results that overall prevalence rate of *Cryptosporidium* in Zoige County was 8.3% (20/242), and three different *Cryptosporidium* species were involved, including *C. bovis*, *C. ryanae*, and an unidentified *Cryptosporidium* sp.; among them, an unidentified *Cryptosporidium* sp. was detected in *O. curzoniae*, and *C. bovis* and *C. ryanae* were detected in *B. grunniens*. These findings demonstrate that *Cryptosporidium* infections are present in both *O. curzoniae* and *B. grunniens* in Zoige County, with notable differences in infection rates and species composition, and provide valuable data on the epidemiology of *Cryptosporidium* in the region and offer scientific support for the healthy development of the livestock industry and public health management.

## 1. Introduction

*Cryptosporidium* was first reported in 1907 [1] and is classified within the Apicomplexa (Phylum), Conoidasida (Class), Eucoccidiorida (Order), Cryptosporidiidae (Family), and *Cryptosporidium* (Genus) [2]. Its life cycle involves oocysts that excyst within the host’s gastrointestinal tract to release sporozoites, which develop through trophozoite–meront–gamont–zygote stages before forming thin- or thick-walled oocysts that are excreted in the feces [3,4]. To date, 49 valid species and over 120 genotypes of *Cryptosporidium* have been recognized, of which approximately 23 have been reported to infect humans, including *C. hominis* [5], *C. meleagridis* [6], *C. canis*, *C. ubiquitum*, *C. cuniculus* [7], *C. felis* [8], and *C. viatorum* [9,10,11]. Surveys indicate that human cryptosporidiosis cases have been documented in more than 90 countries and regions worldwide [12,13]. *Cryptosporidium* parasitizes the gastric or small-intestinal epithelial cells of a wide range of mammals, including livestock, humans, companion animals, and wildlife, and causes cryptosporidiosis, which is characterized by diarrhea in both humans and animals. In poultry and livestock, infection can lead to enteric and respiratory disease, manifesting as diarrhea, weight loss, and growth retardation, thereby posing a serious threat to animal husbandry; no fully effective drugs or vaccines for treatment or prevention currently exist. Vermeulen et al. used the GloWPa-Crypto L1 model to estimate global livestock-derived *Cryptosporidium* oocyst excretion, finding that approximately 3.2 × 10^23^ oocysts are shed annually through livestock feces, with cattle contributing the largest share, followed by chickens and pigs [14]. In China, over 24 provinces, autonomous regions, and municipalities have reported bovine cryptosporidiosis, with infection rates typically exceeding 10% in Shanxi [15], Jiangxi [16], Xinjiang [17], Ningxia [18], Yunnan [19], and Inner Mongolia [20]. Meta-analyses have demonstrated variation in infection prevalence among different cattle populations: Cai’s meta-analysis of dairy cattle in China reported an overall prevalence of 17.0% (3901/33,313) [21], while Geng’s meta-analysis of yak reported a prevalence of 10.52% (1192/8012) [22]. Additionally, infections in domestic dogs and cats are considered important zoonotic reservoirs for human cryptosporidiosis.

The plateau pika (*Ochotona curzoniae* (*O. curzoniae*)) is a small lagomorph in the family Ochotonidae, which is widely distributed across Tibet, Qinghai, southern Gansu, and northwestern Sichuan in China [19]. Despite its high reproductive capacity and abundance, reports of *Cryptosporidium* infection in *O. curzoniae* are extremely scarce; to date, only Zhang has documented such infections on the Qing–Tibetan Plateau [23]. Zoige County, located in northern Sichuan at the northeastern margin of the Tibetan Plateau, is a major pastoral region of northwestern Sichuan, encompassing some 10,620 km^2^ of grassland with an average altitude (3400–3700 m), annual temperature (0–2.0 °C), and annual rainfall (600–700 mm). Yak (*Bos grunniens* (*B. grunniens*)) represents the primary livestock resource and economic base for local herders. Both *O. curzoniae* and *B. grunniens* share the same grassland habitat and are herbivorous species. Due to their prolonged sympatric coexistence, *Cryptosporidium* infection in either species can readily spread to the other, establishing bidirectional transmission and enabling each to serve as a potential reservoir host. Furthermore, local herders’ traditional lifestyle and low awareness of personal hygiene substantially increase the risk of zoonotic disease transmission. However, no studies have assessed *Cryptosporidium* infection in *O. curzoniae* and *B. grunniens* in Zoige County. Thus, this study is the first to investigate their infection status, with the aim of providing data to inform control measures, support healthy livestock development, and enhance public health in the region.

## 2. Materials and Methods

### 2.1. Sample Collection

A total of 114 *O. curzoniae* were captured between March and December 2023, of these *O. curzoniae*, 24, 20, 20, 20, and 30 were captured from Dazhasi, Axi, Hongxing, Maixi, and Tangke, respectively. *O. curzoniaes* were captured using mouse snap traps. Then, to avoid cross-contamination, the intestinal (colon) and gastric contents from each *O*. *curzoniae* were collected separately using sterile plastic gloves, stored in liquid nitrogen, and transported to the laboratory and stored at −80 °C. The body of each *O*. *curzoniae* was deeply buried to avoid being eaten by dogs, cats, or other wild carnivores. When determining the results, if either intestinal (colon) or gastric content tested positive, then *O. curzoniae* was determined as positive. In the same time period, a total of 128 fresh fecal samples from *B. grunniens* (2–3 years old) were collected from Dazhasi (*n* = 18), Axi (*n* = 40), Hongxing (*n* = 26), Maixi (*n* = 14), and Tangke (*n* = 14), located between 102°08′–103°39′ E longitude and 32°56′–34°19′ N latitude.

### 2.2. DNA Extraction

Approximately 150 mg of intestinal and gastric contents of *O. curzoniae* and 200 mg fecal samples of *B. grunniens* were added to ddH_2_O, and homogenates were centrifuged for 5 min at 5000× *g*. The total DNA of all samples was extracted using the TIANamp Genomic DNA Kit (TIANGEN Biotech Co., Ltd., Beijing, China, Cat. No.GDP304, https://en.tiangen.com/ (accessed on 20 December 2023)) according to the manufacturer’s instructions. After each extraction procedure, to assess the quality of the DNA, the DNA concentration was quantified using NanoDrop 2000 (Thermo Scientific, Wilmington, DE, USA), with samples demonstrating concentrations > 30 ng/μL (A260) being selected for subsequent experiments. The extracted genomic DNA was stored in a freezer at −20 °C until PCR amplification.

### 2.3. PCR Amplification

*Cryptosporidium* spp. in the samples was confirmed by nested PCR amplification of the *SSU rRNA* gene according to the previous report [24,25]. A PCR product of approximately 760 bp was first amplified with primers CrF1: 5′-GACATATCATTCAAGTTTCTGACC-3′ and CrR1: 5′-CTGAAGGAGTAAGGAACAACC-3′. A secondary PCR product of about 580 bp was then amplified with primers CrF2: 5′-CCTATCAGCTTTAGACGGTAGG-3′ and CrR2: 5′-TCTAAGAATTTCACCTCTGACTG-3′ [24,25]. The amplification reaction was conducted in a 25 µL volume containing 1.0 µL template DNA (10 U µL^−1^), 12.5 µL 2× PCR mix (TransGen Biotech Co., Ltd., Beijing, China, Cat. No. AS111), 2.0 µL dNTPs, and 9.5 µL distilled water. Each PCR included a positive control (DNA of *C. bovis* preserved in the laboratory) and a negative control (nuclease-free water). The PCR cycling conditions for the *SSU rRNA* gene of *Cryptosporidium* spp. comprised 98 °C for 2 min, followed by 35 cycles of 98 °C for 10 s, 57 °C for 10 s, and 72 °C for 6 s, and then a final extension at 72 °C for 2 min. The PCR products (3 μL) were electrophoresed at 100 V for 25 min on a 1.3% agarose gel, and the results were observed in a gelimager (Tanon Science & Technology Co., Ltd., Shanghai, China, Cat. No. Tanon 1600).

### 2.4. Sequence Analysis and Phylogenetic Tree

The final positive PCR products were subjected to Sanger bidirectional sequencing by Sangon Biotech Company (Chengdu, China) using the second pair of primers. Firstly, all the obtained sequences were analyzed and manually edited by employing DNA Star (https://www.dnastar.com/ (accessed on 12 January 2024)) and were subjected to a nucleotide BLAST (https://blast.ncbi.nlm.nih.gov/Blast.cgi, accessed on 15 January 2024) search through the NCBI database. Subsequently, the sequences with the highest similarity to the blast results were selected (1–4 sequences), and the *SSU rRNA* gene sequences of major *Cryptosporidium* spp., including *C. ryanae*, *C. bovis*, *C. hominis*, *C. meleagridis*, and *C. felis* reported in China, were selected as reference sequences to construct a phylogenetic tree. Lastly, the phylogenetic tree was constructed based on the Neighbor-Joining (NJ) method using MEGA 11.0 (https://www.megasoftware.net/, accessed on 5 March 2024), and 1000 replicates (bootstrap value) were selected to assess the robustness of the findings.

### 2.5. Statistical Analysis

Firstly, the Pearson Chi-square test with the software SPSS 19.0 (IMB Corporation, Armonk, NY, USA) was used to assess whether there was a significant difference in *Cryptosporidium* spp. prevalence between different sampling locations, *O. curzoniae*, and *B. grunniens*. A *p*-value of <0.05 was considered significant. Secondly, the specimen numbers analyzed in the present study are relatively low; therefore, the statistical outcomes obtained were reanalyzed with Fisher’s exact test and confirmed [26]. In addition, the strength of the association between prevalence and test conditions was assessed by calculating the odds ratio (ORs) and 95% confidence interval (95% CI) [27].

## 3. Results

### 3.1. Occurrence of Cryptosporidium *spp.*

The analysis revealed that among the 242 hosts, 20 were positive for *Cryptosporidium*, with an overall prevalence of 8.3%. The infection rates of *O. curzoniae* and *B. grunniens* were 7.0% (8/114) and 9.4% (12/128), respectively (Table 1). There were no significant differences in the prevalence of *Cryptosporidium* spp. between the *O. curzoniae* and *B. grunniens* (χ^2^ = 0.441, *p* = 0.506). In our study, we identified three *Cryptosporidium* species: *Cryptosporidium* sp. (*n =* 8), *C. bovis* (*n* = 10), and *C. ryanae* (*n* = 2). In *O. curzoniae*, only one positive sample was detected in gastric contents, while all other positives were found in intestinal contents. Three novel *Cryptosporidium* spp. genotypes were identified, all exclusively detected in Maixi town, with no occurrences observed in the other four surveyed townships. Among the surveyed townships, *Cryptosporidium* sp. infection rates in *B. grunniens* were highest in Hongxing town (27.9%, 7/26), followed by Maixi town (21.4%, 3/14) and Tangke town (14.3%, 2/14). there were no obvious differences in the prevalence of *Cryptosporidium* spp. among the three regions from *B. grunniens* (χ^2^ = 0.66, *p* = 0.72). For location, Maixi town (32.4%, 11/44) had a significantly higher infection rate than Tangke town (4.6%, 2/44), with statistical significance (χ^2^ = 8.12, *p* = 0.004). However, no statistically significant difference was detected between Hongxing town (15.2%, 7/46) and Maixi town and Tangke town. *C. ryanae* and *C. bovis* were identified only in *B. grunniens*; among them, *C. ryanae* (*n* = 2) was detected only in Hongxing town, while *C. bovis* was detected in Hongxing town (*n* = 5), Maixi town (*n* = 3), and Tangke town (*n* = 2).

### 3.2. Phylogenetic Analysis Based on SSU rRNA Gene of Cryptosporidium

All PCR products of the *SSU rRNA* genes for *Cryptosporidium* spp. were sequenced and compared to each other. A total of five unique sequences were obtained and submitted to GenBank; among these, three sequences from *O. curzoniae* and two from *B. grunniens* were named *Cryptosporidium* sp. Z-OC1 (PP463757), *Cryptosporidium* sp. Z-OC2 (PP463758), *Cryptosporidium* sp. Z-OC3 (PP463759), *C. bovis* Z-Y1 (PV536157), and *C. ryanae* Z-Y2 (PV536158). As shown in Figure 1, *Cryptosporidium* sp. Z-OC1 (PP463757) was clustered with the *Cryptosporidium* sp. isolate from Mongolian pika genotype (OR565220), with the closest genetic relationship and a similarity of 98.3%. As for *Cryptosporidium* sp., Z-OC2 (PP463758) and *Cryptosporidium* sp. Z-OC3 (PP463759) were clustered with the *Cryptosporidium* sp. (KF971356) isolate from *B. grunniens* in Qinghai Province, China, exhibiting 99.17% and 94.60% genetic similarity, respectively; *C. bovis* Z-Y1 (PV536157) was clustered with the *C. bovis* isolate DF1 (PP335809) from *B. grunniens* in Daofu County of Sichuan Province, with 100% similarity; *C. ryanae* Z-Y2 was clustered with the *C. ryanae* (KJ020910) from calves in Shaanxi Province, with 100% similarity.

## 4. Discussion

This is the first report of the detection of *Cryptosporidium* spp. in *O. curzoniae* and *B. grunniens* in Zoige County of Sichuan Province, with occurrences of 7.0% (8/114) and 9.4% (12/128), respectively. Among the eight *SSU rRNA*-positive *O. curzoniae* samples, only one was amplified from gastric contents-designated *Cryptosporidium* sp. Z-OC3 (GenBank accession no. PP463759), whereas the remaining positives originated from intestinal contents. This may be related to the internal environment of the stomach, where a large amount of gastric acid is secreted, which can inactivate most oocysts. Therefore, there are relatively few *Cryptosporidium* species that can parasitize the stomach, mainly *C. muris* and *C. andersoni* [28]. In addition, a study on the infectivity of *Cryptosporidium* in the gastrointestinal tract of Mongolian gerbils showed that there is cross-immunity among gastric *Cryptosporidium* species, but not between intestinal and gastric isolates [29]. Research on *Cryptosporidium* in *O. curzoniae* is exceedingly limited: to date, Zhang (2018) reported a 6.3% (4/64) prevalence in Qinghai *O. curzoniae*, identifying two genotypes–one novel and *C. parvum* [23]. In contrast, lagomorphs more broadly have been studied, such as a meta-analysis by Chen (1951–2024) found a global prevalence of 6.1% in lagomorphs (domestic rabbits, wild hares, and pikas), with pika infection at 17.1% [30]; another meta-analysis of Chinese lagomorphs (1981–2023) reported an overall prevalence of 3.9%, with pika infection at 6.2% [31], suggesting that members of Ochotonidae are particularly susceptible. Our detection rate of 7.0% (8/114) in Zoige County *O. curzoniae* aligns closely with these earlier findings. Notably, all *O. curzoniae* positives in our study derived from Maixi town—the only site with alternating meadow and sandy soils, severe northern desertification, and fragmented grassland—where higher host density likely elevates transmission risk. Additionally, the relatively small sample size may also contribute. Future investigations should include additional herbivores and larger sample numbers to map *Cryptosporidium* distribution across Zoige.

In this study, *B. grunniens* exhibited a 9.4% (12/128) detection rate, lower than reports from Qinghai Province in 2013, 2014, and 2016—24.2% (142/586), 30.0% (98/327), and 28.5% (158/554), respectively [32,33,34], and lower than our laboratory’s earlier finding of 14.8% (32/216) in *B. grunniens* from Hongyuan County, Sichuan [35]. Conversely, our detection rate exceeds those reported in Naqu Prefecture, Tibet (9.6%, 105/1100), and for white yaks in Gansu (5.3%, 4/76) [36,37]. Collectively, these data illustrate regional variation in *B. grunniens Cryptosporidium* prevalence. Additionally, the number of samples, age distribution of samples, grazing method, and sample sizes could be the contributing factors. Cattle are recognized as the most common mammalian host for *Cryptosporidium*, with pre-weaned calves being the principal reservoir. Across 24 Chinese provinces, bovine prevalence is estimated at 14.5% (5265/36,316), rising to 45.8% (141/308) in diarrheic calves [38]. Meta-analyses of various Chinese hosts yield descending overall prevalence of dairy cattle (17.0%; 3901/33,313) [21], swine (12.2%; 4349/30,404) [39], yaks (10.5%; 1192/8012) [22], sheep and goats (4.9%) [40], and lagomorphs (3.9%) [18]. A Nigerian meta-analysis similarly ranked cattle highest at 26.1%, followed by goats (26.0%), pigs (20.0%), sheep (16.6%), humans (15.0%), laboratory animals (9.0%), and birds (7.2%) [41]. These findings underscore the need for enhanced monitoring of *Cryptosporidium* in bovids across the Qinghai–Tibetan Plateau.

Phylogenetic analysis of *SSU rRNA* sequences in this study identified three taxa, *Cryptosporidium* sp., *C. bovis*, and *C. ryanae*, consistent with previous Sichuan reports [33,42]. *Cryptosporidium* sp. was recovered from *O. curzoniae*, whereas *C. bovis* and *C. ryanae* were detected in *B. grunniens*. To date, *B. grunniens* have been found to harbor *C. ryanae*, *C. bovis*, *C. baileyi*, *C. andersoni*, *C. suis-like*, *C. parvum*, *C. hominis*, *C. canis*, *C. struthionis*, *C. ubiquitum*, *C. xiaoi*, *C. suis*, and several novel genotypes [22,32,33,34,35,36,37,43], with *C. ryanae*, *C. bovis*, *C. andersoni*, and *C. parvum* being most common in bovines [44,45]. *C. bovis* was first reported from cattle feces in 2005 by Fayer et al., with a prevalent low pathogenic species mainly infecting post-weaned and adult cattle [46]. It was first reported in Indian dairy farm workers in 2010 [47], and subsequently in Australian farm workers [48], Egyptian children with diarrhea [49], and Colombian individuals [50], indicating zoonotic potential. *C. ryanae*, first detected in cattle in 2008 [51], is a common, low-pathogenic, or asymptomatic bovine species capable of infecting cattle of all ages, with no confirmed human cases to date. Currently, *C. bovis* has been detected in sheep or goats from China [52], the United States [53], Poland [54], Iran [55], and Tunisia [56], and was first identified in camels on the Qinghai–Tibet Plateau in 2020 [57], but has not yet been reported in other animals. In contrast, *C. ryanae* has been identified in horses [58], red deer [59], Tibetan sheep [52], domestic cats [60], marsh deer [61], and brown rats [62], indicating that both *C. bovis* and *C. ryanae* have potential for cross-species transmission. Among them, *C. bovis* is mainly transmitted between cattle and sheep, whereas *C. ryanae* may have a broader host range for cross-species transmission.

In our study, the species of *Cryptosporidium* detected in *O. curzoniae* and *B. grunniens* in Maixi town differed, with a finding similar to those of related studies on the Qinghai–Tibetan Plateau [23,33,63], and likely related to the host specificity of *Cryptosporidium* infections. Research shows that cattle are mainly infected with *C. parvum*, *C. ryanae*, *C. andersoni*, and *C. bovis* [21]; rodents with *C. muris*, *C. tyzzeri*, and *C. viatorum* [64]; sheep with *C. parvum*, *C. xiaoi*, *C. ubiquitum*, and *C. ovis* [65]; and lagomorphs with *C. cuniculus* [30,31]. For example, on the same farm in Qinghai Province, *B. grunniens* were found to harbor *C. andersoni*, *C. bovis*, *C. ryanae* cattle type, *C. ryanae* buffalo type, and *C. suis*-like, whereas Tibetan sheep carried *C. xiaoi* and *C. ubiquitum* [33]. In cattle on the Qinghai–Tibetan Plateau, *C. andersoni*, *C. canis*, *C. bovis*, *C. hominis*, *C. struthionis*, *C. ryanae*, and *C. serpentis* have been detected, while in sheep, *C. parvum* and *C. canis* have been reported [63]. In Qinghai field mice, *C. parvum*, *C. ubiquitum, C. canis*, and a novel genotype were identified, whereas in *O. curzoniae*, only *C. parvum* and a novel genotype were found [23]. According to existing studies and the phylogenetic tree constructed in this study, the possibility of transmission of *C. ryanae* and *C. bovis* between *B. grunniens* and *O. curzoniae* appears to be very low. However, reports on *Cryptosporidium* infection in *O. curzoniae* are currently scarce, and it remains unclear whether other *Cryptosporidium* species (e.g., *C. cuniculus*, *C. parvum*, and *C. ubiquitum*) can be transmitted among *O. curzoniae*, *B. grunniens*, and humans. Therefore, continuous surveillance of *Cryptosporidium* infection in humans, domestic animals, and wildlife in Zoige County is still necessary.

## 5. Conclusions

In summary, for the first time, this study documents *Cryptosporidium* infection in *O. curzoniae* and *B. grunniens* in Zoige County, revealing differences in prevalence and species composition. Given the paucity of epidemiological data on *O. curzoniae* and incomplete knowledge of circulating *Cryptosporidium* species and genotypes, comprehensive investigations are warranted to assess the parasite’s distribution among other livestock and wildlife hosts in the region in the future.

## Figures and Tables

**Figure 1 animals-15-02140-f001:**
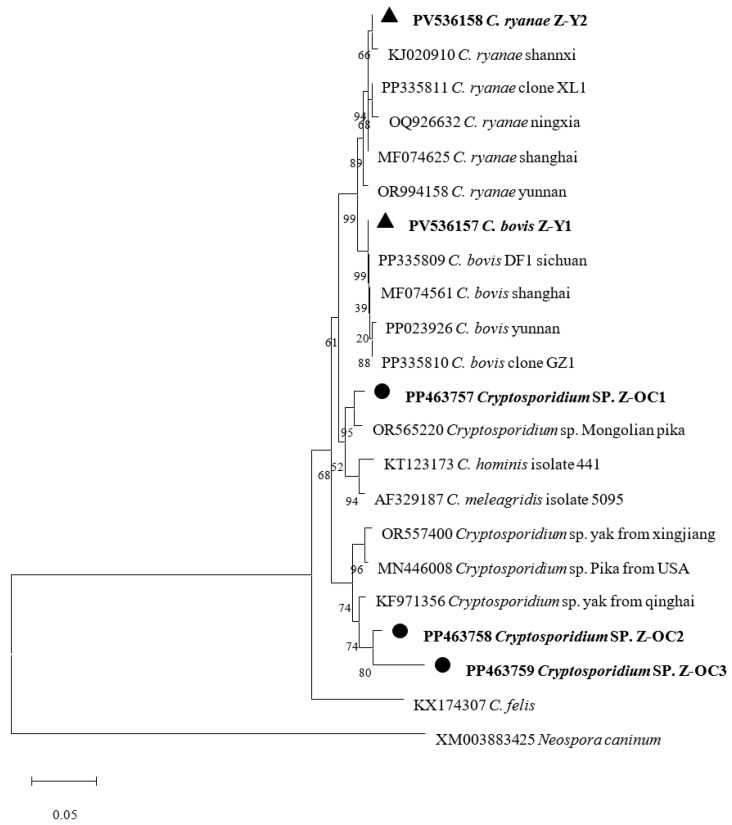
Phylogenetic tree based on *Cryptosporidium SSU rRNA* gene. ●: Sequence obtained from *O. curzoniae*. ▲: Sequence obtained from *B. grunniens*. Bootstrap values from 1000 pseudoreplicates are indicated at the left of the supported node. Scale bar indicates an evolutionary distance of 0.05 substitutions per site in the sequence.

**Table 1 animals-15-02140-t001:** The infection rates of *Cryptosporidium* in *O. curzoniae* and *B. grunniens* in Zoige County of Sichuan Province.

Location	Host	No. Positive/No. of Samples	Prevalence% (95% CI)	OR (95% CI)	*Cryptosporidium* spp.	Location Subtotal % (95% CI), OR
Dazhasi	*O. curzonia* *e*	0/24	0			
	*B. grunniens*	0/18	0			
Axi	*O. curzonia* *e*	0/20	0			
	*B. grunniens*	0/40	0			
Hongxing	*O. curzonia* *e*	0/20	0			15.2 (6.6–28.0), 3.8
	*B. grunniens*	7/26	27.9 (12.3–47.0)		*C. ryanae* (*n* = 2), *C. bovis* (*n* = 5)	
Maixi	*O. curzonia* *e*	8/20	40.0 (19.1–63.9)	2.4 (0.5–11.6)	*Cryptosporidium* sp. (*n* = 8)	32.4 (17.2–51.2), 10.0
	*B. grunniens*	3/14	21.4 (4.8–46.6)	Reference	*C. bovis* (*n* = 3)	
Tangke	*O. curzonia* *e*	0/30	0			4.6% (0.6–14.8), Reference
	*B. grunniens*	2/14	14.3 (1.8–37.8)		*C. bovis* (*n* = 2)	
Host subtotal	*O. curzonia* *e*	8/114	7.0 (2.3–11.7)	Reference		
	*B. grunniens*	12/128	9.4 (4.9–15.7)	1.4 (0.5–3.5)		
Total		20/242	8.3 (5.1–12.5)			

Reference: The reference group used for calculating the odds ratio (OR). The OR value for this group is fixed at 1.0. OR values for other groups represent the risk ratio relative to this group. Reference group for OR calculation, adjusted for location and host. OR: Odds Ratio. CI: Confidence Interval.

## Data Availability

The sequences generated in this study were submitted to GenBank. The names of the repository/repositories and accession number(s) can be found below: https://www.ncbi.nlm.nih.gov/genbank/ (accessed on 18 April 2025), PP463757, PP463758, PP463759, PV536157, and PV536158.

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
