# Peer review of "Molecular Epidemiological Survey of Cryptosporidium in Ochotona curzoniae and Bos grunniens of Zoige County, Sichuan Province"

_animals, 2025, doi:10.3390/ani15142140_

Round 1
Reviewer 1 Report
Comments and Suggestions for Authors
The manuscript on title “Molecular Epidemiological Survey of Cryptosporidium in Ochotona curzoniae and Yaks of Zoige County, Sichuan Province” is an interesting assessment and important research to link parasite overlap between wildlife and domestic animal which may also link to zoonotic scenario in following days. Here there are some feedback that need to address to enhance the manuscript:
Comments
- The manuscript has numerous grammatical, syntactical, and typographical errors that hinder comprehension. A thorough language revision by a native English speaker or professional editor is highly recommended
- Check line 8: "Fcaulty of Agriculture"????
- Check Line 106-107….missing reference?
- Line 115: The PCR cycling condition the SSU rRNA gene of Cryptosporidium spp. comprised..." here Missing preposition (“for”) and awkward phrasing
- Redundant use of “species” with “spp.” ("spp." already implies multiple species)
- there were no significant differences in the prevalence of Cryptosporidium spp. among the O. curzoniae and yak. Using among for two species is not correct
- Line 43: Yaks or yak “"...yak in Zoige County, with notable differences in infection rates..."
- Line 143: "...Hongxing(27.9 %, 7/26)..." space
Lots of similar types or errors are there in manuscript that need thorough revision
- Line 92-96: Explain how the pikas were captured and handled ?
- DNA extraction and PCR protocols (Lines 99–118): More details are needed on quality control measures (e.g., positive/negative controls, contamination avoidance).
- Sample storage (Line 94): The authors mention storage in liquid nitrogen. Clarify duration and whether repeated freeze-thaw cycles were involved.
- The discussion section needs more detail description for robustness of the manuscript
- Line 126-130: The statistical methods are weakly described. No justification is provided for the tests used, and effect sizes or confidence intervals are missing in comparative outcomes
- Revised the Lines 132–134: “20 out of 242 O. curzoniae gastric contents and yak fecal samples were identified as Cryptosporidium spp. positive...”
- Lines 142–144: Check the preposition for location “ during” cannot be used; “the highest infection of Cryptosporidium sp. was recorded during the Hongxing(27.9 %, 7/26)...”
- The discussion section has lack of Depth in Interpretation: The section largely repeats the results rather than offering deeper biological or ecological interpretation. There is insufficient discussion on:
- Zoonotic implications
- Cross-species transmission pathways and,
- Environmental or anthropogenic factors influencing prevalence
- The broad comparison across species in China and Nigeria is informative but, this section lacks focus/objective of study and doesn’t link clearly to the current study focus and area.
Reviewer 2 Report
Comments and Suggestions for Authors
This manuscript have been reported the molecular survey in different areas of Sichuan Province using molecular detection in two host species. The result reflect the role of host harboring the Crytosporidium spp. and showing the difference of positive rate in these study sites. It can be a baseline information for future study due to the pathogen was a zoonotic parasite that can affect to human health or who related the animal hosts. This study will be interested from who in a field of health organization, parasitologists, veterinarians and lecturers or student who studying the protozoan parasites. However, the language and writing style need to be improved. In materials and methods section, it need to be clarify in particular the data analysis whether molecular and statistical methods.
This manuscript have been suggested to revise including:
L19: Please specify O. in O. curzoniae. It should be define the Genus at first mention. Please introduce the animal species in the same style such as common name (scientific name), for the first mention of every sections in text.
L41: please specify how unidentified Crytosporidium sp. do in each host such as "found" or "infected".
L46: epidemiological survey is more specify and consist with the title name. Please consider to replace the "Epidemiology" in keywords.
L51-52 and 56-58: Please insert the citation (s).
L73: Please use the full scientific name "Ochotona curzoniae " and keep common and scientific name style at first mention.
L80: Please add the scientific name in parenthesis with the same style with the plateau pika.
L81: Please use the name in same style. only one between common or scientific name, revising throughout this manuscript.
L107-108: Please cite the original source of primer sets.
L120: Please specify the sequencing methods. Is it the Sanger method. If it is, which primer was used to sequencing.
L122: Please explain how DNA sequences was checked the error of sequencing.
L123: Please give the criteria of DNA sequences using to construct the phylogenetic tree.
L127: Please specify the statistical method to differentiate the sampling location and animal host.
L130: 95% CI can not evaluate the association. Please specify the statistical method to association assessment.
L163: In Table 1, 95% CI was not a prevalence. It mean the confident interval of positive rate in unit of %. Please merge into the Positive rate with 95% CI in parenthesis.
L181: Please provide the Phylogenetic method used, criteria of data set such as how long of DNA fragments and selected selected criteria (host species, geographic distributions or etc.).
L246-247: From Phylogenetic tree of Crytosporidium spp. showed completely separated groups from these two species of animal host. this result can be the answer of this mention about "...unclear...". Please check the relationship between host species that possibly transmitted these two Crytosporidium spp. between different host species.
L280: References less than 50 (only 35), it does not meet the criteria of journal: Animals.
Reviewer 3 Report
Comments and Suggestions for Authors
The manuscript «Molecular Epidemiological Survey of Cryptosporidium in Ochotona 2 curzoniae and Yaks of Zoige County, Sichuan Province» by Tian-cai Tang, Liang-quan Zhu, Ri-hong Jike, Hong-xi Chen, Chen-dong Xiao, Yao Pan, Ke-lei Zhou, Ying-lin Li, Qu-wu Jise, Chao-xi Chen, Li-li Hao presents the first report of Cryptosporidium infection in plateau pikas (Ochotona curzoniae) and yaks in Zoige County, Sichuan Province, China. The study provides novel data on the prevalence and species composition of the parasite in these hosts. The results demonstrate that cryptosporidiosis is present in both O. curzoniae and yak populations in Zoige County, with significant differences in infection rates and species distribution. The findings offer valuable insights into the epidemiology of cryptosporidiosis in the region, supporting livestock industry management and public health strategies.
In general, the studies are well designed and provide new data. The data are presented in one figure and one table, and the quality of the presentation is satisfactory. The text is clearly written and interesting to read. However, several edits are recommended to improve clarity, as outlined below. The work is suitable for publication in the "Animals" journal.
Materials and Methods
Line 94 Please could you provide more information about the collection of intestinal and gastric contents?
Line 100 Which part of the intestines was the sample collected from? Were samples of the intestinal and gastric contents from O. curzoniae and yak combined by location and host? I would recommend that you provide a list of the samples submitted for PCR in a table.
Part of the parasite's life cycle relates to the intestinal cells of the host (oocysts release sporozoites which enter the intestinal cells). Have you considered examining the host's intestinal walls (not just the contents) for parasites?
Line 108 Did you choose the primers yourself? If the primers were selected from an existing source, please specify the reference or database from which they were obtained. If you designed the primers yourself, provide a detailed description of the methodology, including the software or tools used.
Results
Line 132 Were the infection rates of Cryptosporidium calculated separately for intestinal and gastric contents? This question relates to the previous one about Line 100, as it is unclear how the samples were combined.
Line 182 It is necessary to specify what the numbers in the Fig.1 mean.
Discussion
Line 209 How old were the yaks included in your study?
Round 2
Reviewer 1 Report
Comments and Suggestions for Authors
no comments
Reviewer 2 Report
Comments and Suggestions for Authors
This manuscript has been revised by following the suggestion. However, scientific context and writing style of animal name should be rechecked such as "Ochotona curzoniae and Yaks" that mixed between scientific name and common name. It should be revised within the same style as "scientific name and scientific name" or "common name and common name". Please improve throughout the manuscript including text in the table. It was reflect the academic style.
Furthermore, this version of manuscript found text error that need to correct including:
L115 and L120: Please revise the text "Error! Reference source not found." Was it citation?
L169: what "reference" in table1 was meaning? Please give the definition.
Reviewer 3 Report
Comments and Suggestions for Authors
All necessary edits were made by the authors.
Author Response
Thank you very much for taking the time to review this manuscript.